# In Vitro Antitumor and Anti-Inflammatory Activities of *Allium*-Derived Compounds Propyl Propane Thiosulfonate (PTSO) and Propyl Propane Thiosulfinate (PTS)

**DOI:** 10.3390/nu15061363

**Published:** 2023-03-11

**Authors:** Enrique Guillamón, Nuria Mut-Salud, María Jesús Rodríguez-Sojo, Antonio Jesús Ruiz-Malagón, Antonio Cuberos-Escobar, Antonio Martínez-Férez, Alba Rodríguez-Nogales, Julio Gálvez, Alberto Baños

**Affiliations:** 1DMC Research Center, 18620 Granada, Spain; 2Department of Pharmacology, Center for Biomedical Research (CIBM), Instituto de Investigacion Biosanitaria de Granada (ibs.GRANADA), University of Granada, 18071 Granada, Spain; 3Primary Care, Andalusian Health Service District Malaga-Guadalhorce, 29004 Malaga, Spain; 4Chemical Engineering Department, University of Granada, Avenida Fuentenueva s/n, 18071 Granada, Spain; 5CIBER de Enfermedades Hepáticas y Digestivas, Instituto de Salud Carlos III, 28029 Madrid, Spain

**Keywords:** antitumor, anti-inflammatory, *Allium*, organosulfur compounds

## Abstract

Increasing rates of cancer incidence and the side-effects of current chemotherapeutic treatments have led to the research on novel anticancer products based on dietary compounds. The use of *Allium* metabolites and extracts has been proposed to reduce the proliferation of tumor cells by several mechanisms. In this study, we have shown the in vitro anti-proliferative and anti-inflammatory effect of two onion-derived metabolites propyl propane thiosulfinate (PTS) and propyl propane thiosulfonate (PTSO) on several human tumor lines (MCF-7, T-84, A-549, HT-29, Panc-1, Jurkat, PC-3, SW-837, and T1-73). We observed that this effect was related to their ability to induce apoptosis regulated by oxidative stress. In addition, both compounds were also able to reduce the levels of some pro-inflammatory cytokines, such as IL-8, IL-6, and IL-17. Therefore, PTS and PTSO may have a promising role in cancer prevention and/or treatment.

## 1. Introduction

Despite the fact that some recent research has shown that certain risk factors can ease the appearance of some types of cancer, the reasons why some people develop these processes while others do not remain unknown. Among these risk factors, the exposure to chemicals and radiation, age, genetics, lifestyle, or underlying chronic diseases, such as inflammatory disorders, seem to have a prominent role, representing major signaling cues driving the activation of cancer in humans [1]. After cardiovascular diseases, cancer remains the second leading cause of death around the world, with 19.3 million new cases and almost 10.0 million deaths in 2020 [2]. This scenario is projected to be amplified drastically soon; hence, continued searching for more effective chemoprevention and treatment therapies is clearly needed to increase surveillance and to lower the treatment cost for cancer care.

Cancer is caused by the proliferation and progression of abnormal cells. Exposure to carcinogens causes DNA damage and mutations at the cellular level due to the failure in DNA repair mechanisms. The proliferation of the damaged cells causes inflammation of the cells and tissues, finally leading to a tumor formation [3]. One of the mutual traits of cancer is the quick formation of aberrant cells that expand beyond their normal borders, affecting neighboring sections of the body and migrating to other tissues. This process is known as metastasis and is the chief reason for death due to cancer [4].

The advent of modern drug therapies has undeniably improved cancer patients’ cares and lives. However, advanced metastasized cancers remain untreatable, and conventional treatment methods, such as chemotherapy, radiation therapy, or immunotherapy, have major side effects and toxicities [5]. Therefore, the identification and development of novel anticancer products based on natural substances with fewer adverse effects have gained attention in recent years [6]. For example, there is a growing interest in the use of natural immunostimulants in combination with the common therapeutic modalities in the treatment of cancer. The supplementation with these products pretends to improve the immune response against tumors and reduce the suppression effect produced by the chemotherapy [7].

The immune system exhibits the chief function in the defense against infected pathogens and harmful antigens, including tumor cells [8]. Although the induction of an acute immune response plays a cardinal aspect in the detection of and combating tumor cells, it is widely described that the shifting into chronic inflammation, as in case of chronic inflammatory bowel diseases, may also increase the incidence of cancer generation due to the excessive production of inflammatory mediators such as cytokines, reactive oxygen species (ROS), and growth factors. Supporting this, it is well known that chronic inflammation triggers different epigenetic mechanisms that shape the tumor microenvironment, affecting the cell plasticity, differentiation, and polarization of immune cells, promoting the release of ROS and cytokine production [9,10]. In fact, elevated levels of ROS in association with an impaired redox balance are common features of cancer progression and resistance to treatments [11]. In addition, ROS production causes DNA damage, which also contributes to cancer development. It has been estimated that 25% of cancer-causing factors are related to chronic inflammation [3]. Accordingly, the current targets of the treatments include mediators-associated inflammatory pathways and/or oxidant-generating enzymes, as the evaluation of different anti-inflammatory drugs in several clinical trials has revealed that they can play dual roles in inflammation and tumorigenesis. However, some severe problems related to a long-term use of these drugs have been identified [3,9]. Consequently, the search for natural compounds with antiproliferative and/or anti-inflammatory properties, which provides a safe use profile, constitutes a line of work of great interest [12,13,14,15]. Furthermore, the potential synergism of conventional drugs with natural compounds introduces a new aspect to fight cancer, which involves a promising approach to improve the effectiveness of treatment while minimizing the adverse effects associated with chemotherapy [16].

A large number of studies point out that suitable dietary patterns may help to prevent cancer or inhibit tumor development in cancer patients [17]. For instance, some plants are rich in bioactive compounds that possess anticancer and immunomodulatory activities with a low risk of cytotoxicity and side effects. These biologically active plant metabolites are known as “phytochemicals” [18,19]. Although phytochemicals are not associated with nutritional functions, they play a key role as responsible compounds for multiple health benefits. Epidemiological studies, as well as in vivo, in vitro, and clinical trials, have shown the ability of a variety of dietary compounds to reduce the risk of different chronic diseases, including cancer. Many phytochemicals have been reported for their immunomodulatory activities and their uses in treatment of and combating several types of cancer through different mechanisms: the enhancement of the activity of the enzymes involved in the inactivation of carcinogens, the suppression of the growth of cancer cells, or affecting metabolic processes [20,21]. Some examples of phytochemicals with anticancer and immunostimulant properties include curcumin from turmeric, epigallocatechin-3-gallate from green tea, resveratrol from grapes, sulforaphane from broccoli, glucosinolates from cruciferous vegetables, or gingerol from ginger [22]. Among these dietary substances, those derived from alliaceous plants such as garlic, onion, or leek stand out. In general, vegetables from the *Allium* genus contain different reputed bioactive molecules including flavonoids, oligosaccharides, amino acids, selenium, and organosulfur compounds (OSCs) [18,19].

The beneficial properties of OSCs obtained from *Allium* spp., such as antimicrobial, anti-inflammatory, antidiabetic, antioxidant, and immunomodulatory, among others, have been broadly reported [22,23,24,25,26]. In addition, some of these compounds, mainly allylic derivatives from garlic, have been described to display a direct antitumoral effect [11,26] or used as adjuvants in chemotherapy treatment, enhancing the activity of drugs or reducing its side effects [27,28,29]. For instance, the anticancer potential of allicin (e.g., diallyl thiosulfinate), the most important OSC from garlic, has been recently reviewed [30], revealing that it suppresses the growth of different types of tumors. However, this compound is very unstable, and even at optimal processing and storage conditions, easily lead to the spontaneous decomposition of secondary OSCs such as diallyl disulfide (DADS). Several experimental studies have demonstrated that DADS also exhibits anti-tumor activity against many lines of tumor cells, including hematological cancers (leukemia, lymphoma), lung cancer, prostate cancer, or colorectal cancer (CRC) [31]. However, DADS has caused appreciable allergic reactions and toxicity, affecting normal cells too. Thus, the use of these compounds in the prevention and treatment of cancer is limited presently [32].

In contrast, despite the fact that epidemiological studies have confirmed that regular onion consumption reduces the incidence of various forms of cancer as well as other diseases associated with oxidative stress [33], much less is known about the biological activities of OSCs derived from onion. In particular, propyl propane thiosulfinate (PTS), the saturated analogue to allicin, and its oxidized derivative propyl propane thiosulfonate (PTSO) (Figure 1), also present antimicrobial and immunomodulatory activities [34,35], but they are highlighted for their higher stability compared to the OSCs derived from garlic. Therefore, as a first contribution to deep in the potential of PTS and PTSO in the chemotherapy treatment of cancer, the present study aims to characterize the in vitro antiproliferative and anti-inflammatory properties of both compounds and analyze some of their mechanisms of action.

## 2. Materials and Methods

### 2.1. Chemicals and Reagents

PTSO and PTS (93.5% purity) were chemically isolated and provided by DOMCA S.A.U. (Granada, Spain). Both compounds were previously dissolved in dimethyl sulfoxide (DMSO). From these solutions, the different dilutions to be tested were prepared using Dulbecco’s Modified Eagle Medium (DMEM) without fetal bovine serum (FBS) or antibiotics. All reagents were purchased from Sigma-Aldrich Química S.L (Madrid, Spain), unless otherwise stated.

### 2.2. Cell Lines and Culture

MCF-7 (a human breast adenocarcinoma line; ECACC 86012803), T-84 (a human colon carcinoma line; ECACC 88021101), A-549 (a human lung carcinoma line; ECACC 86012804), HT-29 (a human colon adenocarcinoma line; ECACC 91072201), Panc-1 (a human pancreatic cancer line; ECACC 87092802), Jurkat E6.1 (a human leukemia line; ECACC 88042803), PC-3 (a human prostate adenocarcinoma line; ECACC 90112714), and SW-837 (a human rectum adenocarcinoma line; ECACC 91031104) were obtained from the Cell Cultures Unit of the University of Granada (Spain). T1-73 (a human osteosarcoma line; CRL-7943) and hMSCs (human adipose-derived multipotent mesenchymal cells; PCS-500-01) were supplied by the American Type Culture Collection (ATCC). PBMCs (peripheral blood mononuclear cells) were obtained from blood samples of healthy volunteers and provided by the biobank of “Sistema Sanitario Público de Andalucía (SPPA)”. All cell lines were cultured in darkness at 37 °C, with a humidified atmosphere of 5% CO_2_, using DMEM supplemented with 10% FBS, 10 mL/L of penicillin–streptomycin 100×, and 2 mM of L-glutamine, except hMSCs that were supplemented with 20% FBS and without antibiotics, and PBMCs that were cultured with RPMI-1460 medium supplemented with 10% FBS.

### 2.3. In Vitro Antiproliferative Assays

In order to calculate the half-maximal inhibitory concentration (IC_50_) values of PTS and PTSO, adherent cells (MCF-7, T-84, A-549, HT-29, Panc-1, SW-837, PC-3, and T1-73) were seeded in sterile 96-well plates (Thermo Fisher Scientific, Denmark) at a high density (1.4 × 10^4^ cells/well) and incubated at 37 °C with 5% CO_2_ for 24 h to allow for cell adhesion. In non-adherent cells (Jurkat and PBMCs), the induction was conducted directly. Increasing concentrations of PTS and PTSO (1–250 μM) were added in the corresponding wells and incubated for 72 h at 37 °C with 5% CO_2_. The effect of both compounds on adherent cell lines was evaluated using a colorimetric technique with Sulforhodamine-B (SRB) [36]. Non-adherent cell lines were quantified by the MTT assay [37]. The optical density values of adherent and non-adherent cells were determined by colorimetry at 490 nm using a microplate reader (Multiskan EX, Thermo Electron Corporation). The assessment of absorbance was obtained using the SkanIt RE 5.0 for Windows v.2.6 (Thermo Labsystems, USA) and a regression analysis for each cell line using Statgraphics 18 software (Statistical Graphics Corp, 2000, Warrenton, VA, USA) was conducted. The IC_50_ values were calculated from the semi-logarithmic dose–response curve by linear interpolation. Finally, the therapeutic index (TI) of each compound was determined to determine the margin of safety of PTS and PTSO when used as an antiproliferative. TI was calculated by establishing the ratio between the IC_50_ values obtained in non-tumoral cells and in a tumor cell line. For each cell line, the assays were performed in duplicate.

### 2.4. Oxidative Stress Assays

MCF-7 and T-84 cells were seeded in 96 well-plates at a high density in sextuplicate. At 24 h, the cells were induced with increasing concentrations of PTS and PTSO, with or without 5mM of NAC (N-Acetyl-Cystein) for 1 h pre-induction. After 72 h, the cell viability was evaluated using the SRB method. The IC_50_ values were calculated from the semilogarithmic dose–response curve by linear interpolation. Assays were performed in duplicate.

The production of intracellular ROS was detected by fluorescence microscopy. MCF-7 cells were seeded at 5 × 10^3^ cells/well for 24 h on a µ-Slide 8 well high glass bottom (Ref.80807, Ibidi, Gräfelfing, Germany). Then, the cells were preincubated in the presence or absence of 5mM of NAC for 1 h and treated with PTS or PTSO for 24 h at doses of IC_50_. Subsequently, the cells were incubated with 2,7-dichlorofluorescein diacetate (DCFH-DA) (10 μM) in darkness at 37 °C for 30 min. Fluorescent images were taken with a confocal laser scanning microscope Confocal Leica TCS-SP5 (Leica, Munich, Germany) at 25× of magnification and 1.5× of zoom.

### 2.5. Apoptosis Assays

The cell viability was determined by flow cytometry using the Annexin V-FITC kit (Trevigen, Gaithersburg, MD, USA). MCF-7 and T-84 cells were seeded at a high density (2 × 10^5^ cells/cm^2^) in 6-well plates. After 24 h, the cells were induced with PTS and PTSO for 48 h at the IC_50_ concentration for each cell line. The cells were detached with the TrypLE Express Enzyme (ThermoFisher Scientific, Waltham, MA, USA), washed with PBS, and collected by centrifugation at 300× *g* for 10 min. Then, the cells were washed again and incubated with annexin-V FITC and propidium iodide (PI) in an annexin-V binding buffer for 15 min. After incubation, the cells were diluted with the binding buffer and examined immediately in a FACScan flow cytometer, using FlowJo (v.7.6.5, Tree Star, Inc., Ashland, OR, USA). This assay was performed in duplicate.

### 2.6. In Vitro Anti-Inflammatory Assays

The HT-29 and T-84 cells were seeded at a high density (1.4 × 10^4^ cells/well) in 96-well plates for 24 h. Then, the supernatants were discarded and the compound dissolved in the supplemented medium was added. After 1 h of incubation with PTS or PTSO, 1 µg/mL of lipopolysaccharide from *Salmonella enterica* serotype *typhimurium* (LPS) was added, and the plates were incubated for 24 h at 37 °C and 5% CO_2_. All the concentrations tested were performed in sextuplicate. After induction, the supernatants were collected, centrifuged at 1000× *g* for 10 min, and stored at −80 °C. Finally, the IL-8, IL-6, and IL-17 determination was carried out by an ELISA using cytokines commercial kits (Invitrogen-ThermoFisher Scientific, Bethlehem, PA, USA). The assays were performed in duplicate.

### 2.7. Statistical Analysis

The results of absorbance in cytotoxicity assays and IC_50_ were evaluated using an analysis of variance (ANOVA), using the statistical software SPSS 11.5 (IBM, New York, NY, USA). All the results were expressed as mean ± standard deviations (SD). Figures and statistical analysis for apoptosis assays and anti-inflammatory assays were generated with GraphPad prism 8.0 software (GraphPad Software Inc., La Jolla, CA, USA) using a one-way ANOVA test supplemented with Tukey’s post hoc. Differences were considered statistically significant when *p* < 0.05. The relative fluorescence intensity was quantified using the software Image J (v. 1.53t).

## 3. Results

### 3.1. In Vitro Antiproliferative Effects of PTS and PTSO

The antiproliferative activity of PTS and PTSO was evaluated in all the cell lines described above. Both compounds inhibited cellular proliferation in a dose–response manner with a different efficacy against the cell lines used (Table 1). The results revealed that the IC_50_ values from PTSO were higher than for PTS, except in MCF-7, Jurkat, SW-837, and Panc-1 (Table 1). Therefore, these findings indicated a different but remarkable antitumor effectiveness of PTS and PTSO (Table 1).

In order to determine the in vitro TI of PTS and PTSO, their effect by performing cultures with PBMCs was studied under the conditions described in Section 2.2. In this cell line, the results obtained show an IC_50_ value of 229.2 µM for PTS and 248.5 µM for PTSO. Therefore, the TIs for PTSO and PTS were 12.9 and 36, respectively, taking as reference the MCF-7 line, and 14.6 and 23.4 considering the Jurkat line.

To confirm the harmlessness of PTS and PTSO in healthy cells, their effect was also tested on hMSCs. It was found that both compounds hardly produced toxicity on these cells as very high concentrations are required to affect cell viability (Figure 2). Moreover, certain concentrations of these compounds even were able to induce the proliferation of hMSCs. Specifically, induction with 10 µM of PTS or PTSO increased the population by 17% and almost 11%, respectively, compared to the control. At concentrations greater than 20 µM, it was noted that PTSO was generally less harmful on hMSCs than PTS.

### 3.2. Oxidative Stress Assays

In order to determine if the mechanism of action of PTS and PTSO was related to ROS production, their cytotoxicity was tested in MCF-7 and T-84 cells in the presence or absence of NAC 5 mM. In MCF-7 cells, the IC_50_ of PTS and PTSO in the presence of NAC barely affected the cell viability, reducing the population by 10% compared to controls without NAC (Figure 3A). Thus, the presence of NAC decreased the anti-proliferative effect of both compounds, increasing by 40% the cell population. In T-84 cells, the IC_50_ of PTS and PTSO in the presence of NAC were able to reduce the population by 20% and 15%, respectively, increasing the cell viability by 33–34% compared to the controls without NAC. From these results, it may be concluded that NAC protects tumor cells from the activity of PTS and PTSO.

These results were also confirmed by the determination of intracellular ROS by a confocal microscope using DCFH-DA in the MCF-7 line. Thus, the cells were treated with PTS and PTSO at IC_50_ concentrations, in the absence or presence of NAC 5 mM. As it can be observed in Figure 3B. NAC incubation reduced the high fluorescence of tumor control cells. Remarkably, the incubation for 24 h only with PTS or PTSO in MCF-7 cells diminished considerably the cell population in the absence of the antioxidant, and so did the fluorescence (Figure 3B). On the contrary, when MCF-7 cells were incubated with PTS or PTSO and previously supplemented with NAC, the findings revealed a higher fluorescence because the population had barely been affected (Figure 3B).

### 3.3. Study of Apoptosis

Cultures of MCF-7 and T-84 cells were incubated with the IC_50_ concentrations of PTS and PTSO for 48 h. The apoptosis induction was assessed by an annexin V FITC assay using flow cytometry. Figure 4A shows the different apoptotic stages in both cell lines when they were incubated with both compounds. Specifically, the percentage of the different apoptotic stages was quantified, and the results showed that in MCF-7 cells, the fraction of early apoptosis increased from 12.7% to 20.2% in cultures treated with PTS, and to 17.3% with those treated with PTSO. The number of late apoptotic cells also increased in treated cells, from 1.6% to 10.7% with PTS and to 5.9% with PTSO (Figure 4B). In T-84 cells, the percentages of early and late apoptotic cells were also higher in treated cells compared to the control for both compounds, although the induction of apoptosis was more evident with PTS, increasing from 0.9% to 7.4% of early apoptotic and 0.3% to 2.8% of late apoptotic cells (Figure 4B).

### 3.4. Evaluation of Anti-Inflammatory Properties

To evaluate the anti-inflammatory properties of PTS and PTSO, the production of IL-8, IL-6, and IL-17 was determined in HT-29 and T-84 cells after their incubation with LPS, which has already been long demonstrated to be capable of eliciting responses associated with inflammation in vitro, including the production of pro-inflammatory cytokines [38]. The concentrations of PTS and PTSO tested were selected considering the levels of cytotoxicity already determined in these lines considering their IC50. Both PTS and PTSO were able to significantly inhibit the LPS-activated production of these cytokines (Figure 5). However, no concentration–response relationship was observed since most of the concentrations assayed showed a similar efficacy for both compounds with some exceptions: when the production of IL-8 in HT-29 cells was considered, the most effective concentrations were 1 μM of PTS and 10 μM of PTSO. (Figure 5A); or when the production of IL-17 was considered in HT-29 and T-84 cells, both compounds showed more efficacy at the highest doses assayed (10 µM and 25 µM) (Figure 5C).

## 4. Discussion

Treatment with the extracts or compounds derived from *Allium* has been the subject of numerous studies and trials to establish a link with a reduced risk of cancer. In this sense, our findings are in concordance with other assays previously published [39,40,41]. In fact, several in vitro and in vivo studies have shown the potential antiproliferative activity of the extracts or compounds derived from *Allium* in the same cell lines we have tested. For instance, in one study conducted with quercetin from *Allium cepa*, this compound showed cytotoxicity against MCF-7, HT-29, PC-3, and Jurkat cells [42]. The antitumor capacity of crude thiosulfinates from *Allium tuberosum* affected the viability of MCF-7 breast tumor cells, with an IC_50_ of 155.1 μM in the case of S-methyl methanethiosulfonate and 51.1 μM for S-methyl 2-propene-1-thiosulfinate [43]. In another study conducted with 22 stabilized thiosulfinates derived from *Allium* vegetables, the IC_50_ of the compound with the greatest anticancer activity in MCF-7 cells (S-4-methoxyphenyl 4-methoxybenzenesulfinothioate) was 46.5 µM [20]. Despite the fact that in this article, PTS was synthesized to carry out studies on the mechanism of action, the IC_50_ in the MCF-7 line was not reported [20]. In our assays, both PTS and PTSO achieved lower IC_50_ values in MCF-7 (17.7 and 6.9 μM, respectively) than the mentioned compounds. Other *Allium* OSCs, whose IC_50_ values at 72 h in MCF-7 have been reported, are allicin (10 μM) [44] and DADS (4.1 μM) [45], showing an antiproliferative effect in this line similar or a little higher than the one obtained in our assays with PTS and PTSO.

Some of the OSCs more commonly studied, as allicin, DADS, or diallyl trisulfide (DATS), have also showed antitumor activity in colon cancer cell lines, including HT-29 [11,45,46,47]. A similar effect has been described for water-soluble garlic-derivatives, such as S-allylmercaptocysteine. The effect of this compound on cell cycle progression and proliferation was evaluated in colon cancer cell lines SW-480 and HT-29, achieving the growth inhibition of both lines inducing apoptosis [48]. Our results showed that PTS and PTSO exerted cytotoxicity in all of the colon tumor cells challenged, being especially remarkable for PTS in T-84 and HT-29, with IC_50_ values below 20 μM (18.2 μM and 15.6 μM, respectively). Conversely, in SW-837 cells, the IC_50_ of PTS was higher than for PTSO (150.8 μM and 132.8 μM, respectively).

Regarding lung tumor cell lines, DADS and DATS have also shown antitumor activity against A-549 by inducing apoptosis [49], though in this study, the IC_50_ was not indicated. In another in vitro assay, DADS (15–120 μM) was tested in AML HL-60 leukemic cells, succeeding in suppressing cell growth [50]. These results are in accordance with those obtained in our experiments since the IC_50_ of PTS and PTSO against Jurkat cells were in the same concentration range (10.6 and 15.7 μM, respectively).

In recent years, the use of blood cells from healthy volunteers has become a model increasingly popular as a method to determine the toxicity of a compound, instead of using established human cell lines [51]. In the assay to test the antiproliferative effect of PTS and PTSO in PBMCs, high concentrations of both compounds were necessary to affect the viability of these healthy cells (IC_50_ > 200 µM). These concentrations lead to high TIs. Another proof of the harmlessness of PTS and PTSO is the fact that certain concentrations of these compounds could increase the population of hMSCs, as seen in Figure 2. Consequently, our results are indicative of the large margin of safety of PTS and PTSO and, therefore, of their potential to be tested in in vivo treatments against neoplastic pathologies. These results in PBMCs are consistent with those obtained in previously conducted in vivo assays, in which it was demonstrated that PTSO did not cause toxicity in Sprague Dawley rats administered 55 mg PTSO/kg body weight/day for 90 days, without showing liver damage, neither clinical signs nor mortality [27,52,53].

Oxidative stress can damage membrane lipids, proteins, and nuclear and mitochondrial DNA in cells. The assays related to oxidative stress revealed that populations of MCF-7 and T-84 decreased significantly after treatment only with PTS or PTSO at IC_50_ concentrations, compared to cells treated with the same concentrations but also pre-incubated with NAC. This effect was more evident in MCF-7 cells, whose population increased around 40% compared to cells induced in the absence of NAC but only with both onion-derivative compounds (Figure 3). These findings may be justified by the fact that NAC exerts an antioxidant and protective effect against PTS and PTSO, which involved lower cell death. In the assay performed with DCFH-DA, a ROS indicator, it was observed that MCF-7 cells incubated only with PTS or PTSO showed a lower fluorescence, which corresponds to a lower cell density (Figure 4). As it is widely known, tumor cells have a high level of oxidative stress compared to healthy cells, and this is related to an increase in ROS production due to changes in their metabolism [54,55,56]. When MCF-7 cells were induced with any of the compounds for 24 h but also preincubated with NAC, their viability was hardly affected as their fluorescence increased. Therefore, this assay confirms the results obtained in the proliferation assays conducted in the absence or presence of NAC (Figure 3). It could be concluded, hence, that oxidative stress seems to be involved in the mechanism of action of PTS and PTSO, as occurs with other known antitumor agents such as elesclomol or paclitaxel, among others [57,58].

Nevertheless, it must be considered that the cytotoxic activity of both compounds could also be due to their pro-apoptotic action. Similar findings have been reported with DADS in experiments conducted with A-549 and PC-3 cells [59], where the treatment with NAC was able to block both the production of ROS (e.g., H_2_O_2_) and apoptosis. Other authors have reported that DATS-induced apoptosis was associated with ROS production in several of the lines tested in our trials, such as MCF-7 [60,61]. However, there are other articles reporting that ROS generation appears to play only a secondary role in the cytotoxicity of OSCs in tumor cells. For example, in an assay conducted with esophageal cancer cells WHCO1 [62], it was observed that ROS was not the main cause of cytotoxicity of garlic-related disulfides, although NAC was still able to interfere with the assay. As oxidative stress and ROS levels seem to affect cancer development, personalized treatments for patients should be addressed, considering the basal antioxidant status, type of cancer, and mechanisms of action of drugs [63,64,65]. Moreover, various studies suggest that the intake of supplements or foods with an antioxidant capacity may not be generally recommended during chemotherapy treatments [66,67].

As previously stated, tumor cells used to be more sensitive to drugs that generate large amounts of ROS, or that affect the ability of cells to eliminate them, which has been associated with their death by apoptosis [68,69]. Apoptosis is the programmed cell death characterized by a series of morphological events, including DNA fragmentation, cell shrinkage, and the formation of membrane-bound apoptotic bodies that are rapidly phagocytized by neighboring cells [70,71]. Our results revealed that there were significant differences in the fraction of early and late apoptosis in MCF-7 and T-84 cells induced with PTS and PTSO, indicating that both compounds would be able to induce apoptosis in tumor cells.

According to the literature, there are studies that state that the regular intake of garlic reduces neoplastic growth and tumor cells proliferation by inducing apoptosis [72,73]. Regarding, specifically, OSCs, it has been reported that DADS showed a significant induction of apoptosis in a human gastric adenocarcinoma cell line [74], and allicin supplementation induced death by apoptosis in several colon tumor cell lines, including HT-29 [75]. Therefore, the way that PTS and PTSO exhibit their antitumor activity seems to be similar to other OSCs. However, since the autophagy and apoptosis regulated by ROS are cellular processes that can interact with each other [76], further studies would be necessary to determine if autophagy is also involved in the mechanism of action of PTS and PTSO.

In summary, the reported IC_50_ of PTS and PTSO in the tumor lines tested are in the same range than those of the common OSCs whose antitumor effect has been proven. Nevertheless, given their higher stability compared to substances such as allicin or DADS, PTS and PTSO could be considered promising candidates to use in anticancer treatments, alone or as adjuvants of chemotherapy drugs. This approach was described by Perez-Ortiz et al., who co-administered a thiosulfinate-enriched garlic extract with 5-fluorouracil (5-FU), achieving a greater effectiveness than standard chemotherapy with 5-FU and oxaliplatin [29]. Similarly, other dietary compounds have also been used as adjuvants, such as curcumin [77], epigallocatechin gallate (EGCG) [78], and lycopene [79].

In the anti-inflammatory assays, both PTS and PTSO were able to reduce the levels of three pro-inflammatory cytokines usually involved in the development of cancer: IL-8, IL-6, and IL-17. Some authors correlate IL-6 levels with tumor stage, the metastasis survival rate, or apoptosis in various types of cancer, such as breast [80] or colon [81], while the production of IL-8 has been linked to pro-tumorigenic roles which influence the tumor microenvironment [82]. The IL-17 cytokine is widely recognized for its ability to modulate the inflammatory response, contributing to the development of chronic inflammation [83], and its level could increase in the serum and tissues of patients with CRC [84]. In fact, in vivo studies related to this type of cancer have shown that IL-17 plays an important role in its prognosis and metastasis [85]. As previously stated, the production of this cytokine and IL-8 was significantly reduced by PTS and PTSO in both cell lines, HT-29 and T-84. However, compared to the control, the reduction of IL-6 was only achieved in HT-29 cells.

Interestingly, most of the highest reductions in the production of pro-inflammatory cytokines were obtained with the lowest concentrations of PTS and PTSO. Thus, these OSCs would not act in a dose-dependent manner to exert their anti-inflammatory activity, making their action dependent on the cancer cell line characteristics. In agreement with our findings, PTS and PTSO have previously demonstrated their immunomodulatory effect in several animal models. Concretely, PTSO was tested in two experimental models of colitis, which were associated with the regulation of cytokines in inflamed colonic tissue, leading to a reduction of pro-inflammatory cytokines IL-1β, TNF-α, and IL-6 [86]. In a more recent work, PTSO showed its capacity of attenuating the obesity-associated systemic inflammation, reducing the expression of the mentioned cytokines in adipose and hepatic tissues in mice [87]. Moreover, in a murine model, PTS was able to normalize the levels of IL-22 of animals fed an obesogenic diet [88].

## 5. Conclusions

PTS and PTSO were able to inhibit the growth of human tumor lines MCF-7, T-84, A-549, HT-29, Panc-1, Jurkat, PC-3, SW-837, and T1-73. In addition, both compounds showed high TIs and were able to induce hMSCs proliferation at low concentrations. Furthermore, PTS and PTSO reduced the values of pro-inflammatory cytokines IL-6, IL-8, and IL-17 in HT-29 and T-84 lines. The generation of ROS and apoptosis seems to be related to the antiproliferative and anti-inflammatory activity of PTS and PTSO in tumor cells. This work represents a promising new therapeutic application of these compounds, although further investigation is needed to deepen the knowledge on the mechanisms of action and demonstrate their efficacy in vivo.

## Figures and Tables

**Figure 1 nutrients-15-01363-f001:**
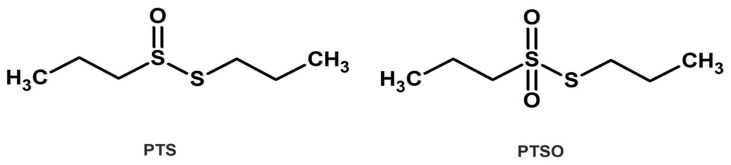
Chemical structure of PTS and PTSO.

**Figure 2 nutrients-15-01363-f002:**
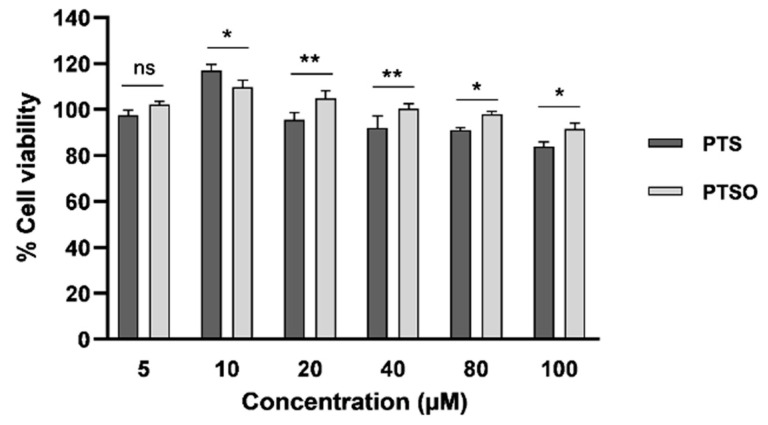
Effects of increasing concentrations of PTS and PTSO on hMSCs viability three days post-induction. The histogram depicts means ± SD of six determinations. * *p* < 0.05; ** *p* < 0.01; ns = non-significant.

**Figure 3 nutrients-15-01363-f003:**
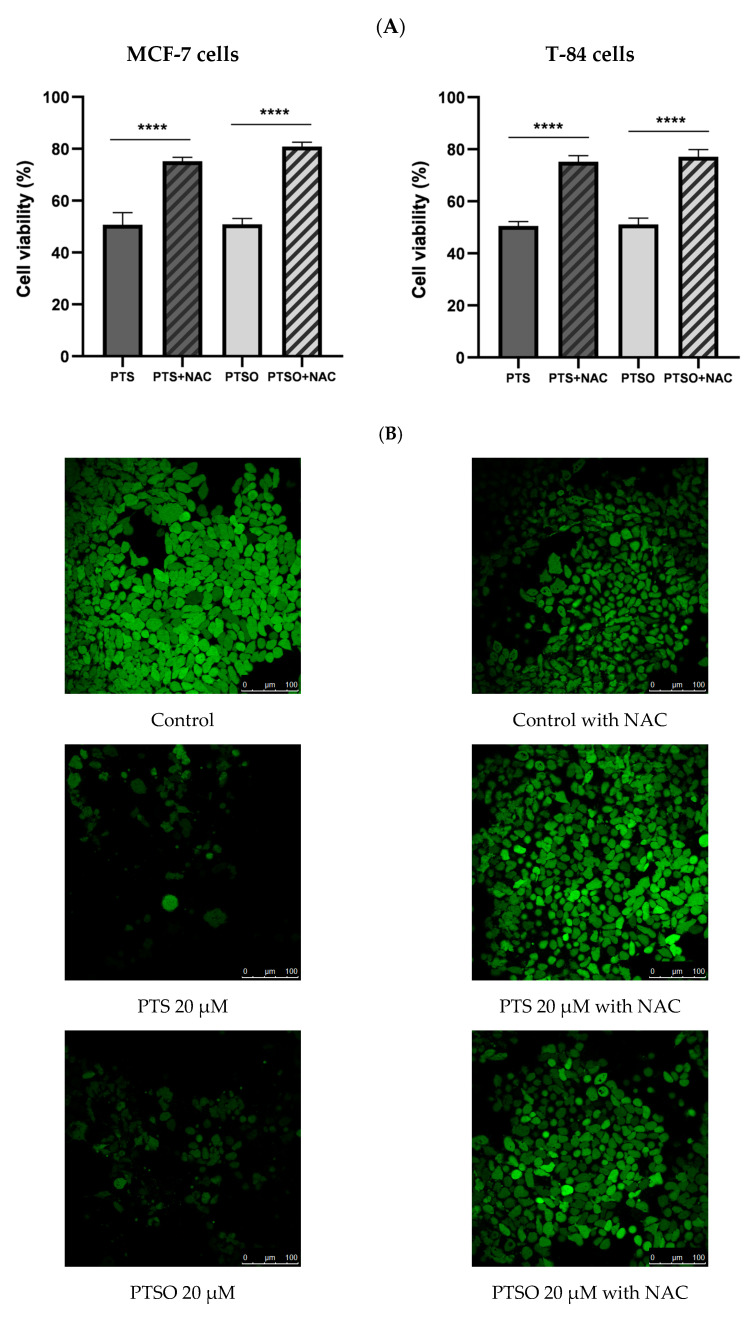
(**A**) Cytotoxicity of PTS and PTSO on MCF-7 and T-84 cells in absence or presence of NAC 5 mM. The concentration of PTS and PTSO was equal to IC_50_ and all cells were induced for 72 h. The histogram depicts means ± SD of six determinations. **** = *p* < 0.0001. (**B**). Effect of PTS and PTSO on ROS production detected by DCFH-DA staining (magnification 25× and zoom 1.5×) in MCF-7 cells. The relative fluorescence intensity was expressed as the mean ± SD. Different letter between columns indicates significant differences at *p* < 0.05.

**Figure 4 nutrients-15-01363-f004:**
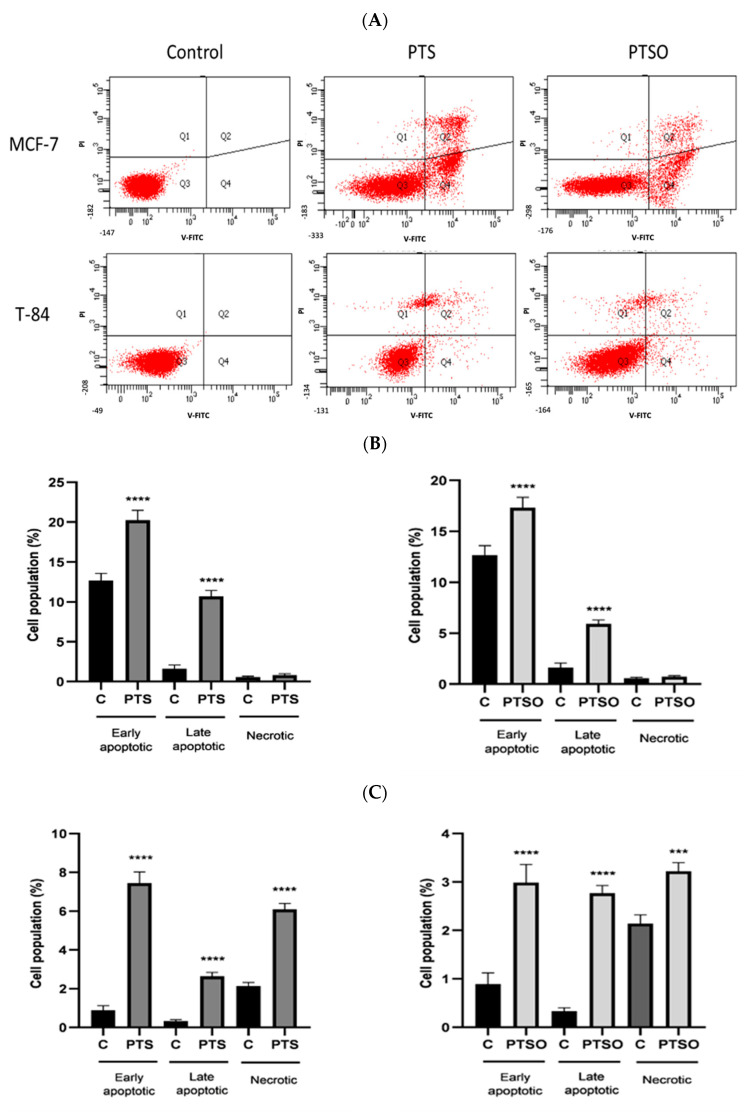
Analysis of apoptosis in MCF-7 (**A**) and T-84 (**B**) cell viability with and without PTS or PTSO for 48 h, revealed with annexin V-FITC/PI staining by cytometry. The graphics show the percentages of early apoptotic, late apoptotic, and necrotic cells after induction with PTS and PTSO at a concentration of IC50 in MCF-7 cells. (**C**) Histograms of MCF-7 and T-84 cells. Control = untreated control cells. The experiment was repeated independently three times yielding similar results. **** = *p* < 0.0001; *** = *p* < 0.001.

**Figure 5 nutrients-15-01363-f005:**
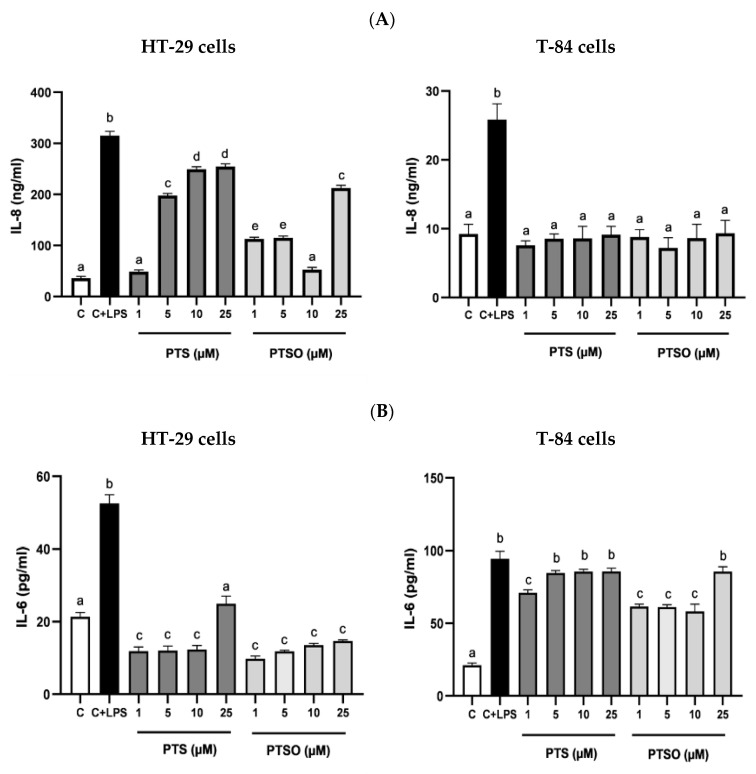
Effect of PTS and PTSO on inflammatory status. (**A**) Impact of PTS and PTSO on IL-8 concentration in HT-29 cells (a) and T-84 cells (b) induced with LPS. (**B**) Effect of both treatments on IL-6 concentration in HT-29 cells (a) and T-84 cells (b) induced with LPS. Cells were induced during 24 h with LPS (1 µg/mL) and increasing concentrations of PTS and PTSO (1–25 µM). (**C**) Evaluation of PTS and PTSO on IL-17 concentration in HT-29 cells and T-84 cells induced with LPS. C = cells not induced with LPS (control), C + LPS: cells induced with only LPS (1 µg/mL). Different letters between columns indicate significant differences at *p* < 0.05.

**Table 1 nutrients-15-01363-t001:** Antiproliferative activity of PTS and PTSO against MCF-7, T-84, A-549, HT-29, Panc-1, Jurkat, SW-837, PC-3, T1-73, and PBMCs 72 h post-induction. Different letters differ significantly (*p* < 0.05).

Cell Line	IC_50_ PTS (µM)	IC_50_ PTSO (µM)
MCF-7 (human breast adenocarcinoma)	17.7 ± 1.9 ^a^	6.9 ± 0.7 ^b^
T-84 (human colorectal carcinoma line)	18.2 ± 2.2 ^a^	37.3 ± 0.8 ^c^
A-549 (human lung adenocarcinoma line)	10.4 ± 1.2 ^d^	38.6 ± 1.1 ^c^
HT-29 (human grade II colorectal adenocarcinoma line)	15.6 ± 2.5 ^a^	50.8 ± 3.1 ^e^
Panc-1 (human pancreatic carcinoma line)	34.5 ± 3.7 ^c^	33.8 ± 4.2 ^c^
Jurkat (human tumor T lymphocytes line)	15.7 ± 1.4 ^a^	10.6 ± 1.3 ^d^
SW-837 (human rectum adenocarcinoma tumor line)	150.8 ± 2.4 ^f^	132.8 ± 1.7 ^g^
PC-3 (human prostate adenocarcinoma tumor line)	128.5 ± 2.3 ^g^	198.7 ± 3.5 ^h^
T1-73 (human osteosarcoma tumor line)	76.4 ± 3.2 ^i^	98.2 ± 2.2 ^j^
PBMCs (Peripheral blood mononuclear cells)	229.2 ± 3.7 ^h,k^	248.5 ± 3.6 ^k^

## Data Availability

Data will be available to be shared upon publication by correspondence with either Alba Rodriguez Nogales (albarn@ugr.es) or Julio Galvez (jgalvez@ugr.es), after approval of a proposal, with a signed access agreement, and relevant ethics consent.

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
