# Peer review of "In Vitro Antitumor and Anti-Inflammatory Activities of Allium-Derived Compounds Propyl Propane Thiosulfonate (PTSO) and Propyl Propane Thiosulfinate (PTS)"

_nutrients, 2023, doi:10.3390/nu15061363_

Round 1

Reviewer 1 Report

The authors identified the efficacy of two sulfur compounds. However, the specific mechanism of action was not revealed and only the phenotype was shown. It is inappropriate for publication in nutrient.

major comments

1.Rather than presenting various biomarkers in an enumerated manner, in-depth research is needed.

2. The effects of these substances on the cell signaling pathway should be shown.

minor comments

1. Making up to figure 10 is too much. It is easier for readers to see if you put several data in one figure and keep it to 5 or less.

2. It would be nice to put the facs PI annexin5 figure in fig.7.

3. Fig.5 and fig3 data also need to be quantified.

4. In fig 9 and fig 10, the authors have to find concentration dependent intervals.

Author Response

Reviewer#1. Comments and Suggestions for Authors

The authors identified the efficacy of two sulfur compounds. However, the specific mechanism of action was not revealed and only the phenotype was shown. It is inappropriate for publication in nutrients.

Major comments

1.Rather than presenting various biomarkers in an enumerated manner, in-depth research is needed.

  1. The effects of these substances on the cell signaling pathway should be shown.

Response: We thank the comments made by the reviewer. Although it is evident that more precise knowledge about the mechanism of action involved in the effects of both compounds through the evaluation of cell signaling pathways would provide valuable information, this challenging aim is too vast and outside the scope of this study. However, we are continuing the studies to get a thorough comprehension of the compounds.

In the present study, we focused on determining the antiproliferative properties of these compounds, tested in different types of cancer cells to explore different mechanisms as well as in non-cancer cells to exclude direct toxic effects, thus supporting the safety profile of these compounds. Additionally, we evaluated their effect on apoptosis and the inflammatory response, two processes that are clearly associated with cancer development. In this regard, the mechanisms behind the effects were since we studied the ability of the products to induce DNA damage, probably related to the release of reactive oxygen species, which can contribute to the antiproliferative effects observed.

Although this study can be initially considered as descriptive, it can establish the basis for future studies that explore in depth the precise mechanism of action and the cellular pathways involved that have been proposed here. As commented above, we are planning these experiments, but these will be performed in specific cancer cell types, mainly in those that have shown a higher sensibility from the results obtained in the present study.

We also think that this study is relevant for Nutrients since it supports the role of dietary interventions with these compounds to prevent or ameliorate cancer development, considering the limitations of a preliminary study in cell lines, and that we should further investigate other mechanisms involved,  the effects in vivo in animal models and in humans at a later stage.

Minor comments

  1. Making up to figure 10 is too much. It is easier for readers to see if you put several data in one figure and keep it to 5 or less.

Response: Following the reviewer suggestion, the figures have been reorganized and the number of figures have been reduced.

  1. It would be nice to put the facs PI annexin5 figure in fig.7.

Response: According to the reviewer comment,  the figures have been modified. 

  1. Fig.5 and fig3 data also need to be quantified.

Response: Following the reviewer suggestion, the fluorescence intensity has been quantified. 

  1. In fig 9 and fig 10, the authors have to find concentration dependent intervals.

Response: In the revised version manuscript, a comment has been included about the lack of the concentration-response relationship when PTS and PTSO were assayed for the antiinflammatory properties, indicating the minimum concentration that has an inhibitory effect. Interestingly, the compounds have shown antiinflammatory activity at all concentrations evaluated. 

Reviewer 2 Report

This manuscript is focused on anti-proliferative, pro-apoptotic and anti-inflammatory activities of two OSCs obtained by Allium spp, PTSO and PTS,  in several tumor cell cultures.  The in vitro experiments revealed that these bioactive compounds can reduce cell proliferation and induce apoptosis and indicate ROS production as one of potential mechanism underling their cytotoxicity effect. Furthermore, the anti-inflammatory effect was investigated by assayed the pro-inflammatory cytokine production.  The topics are well-presented, the language is clear and the text is well structured. The novelty of the work is interesting and the rational is clear, but there are several limitations in the results, then the manuscript is still a long way from publication.

COMMENTS FOR THE AUTHOR:

Major points:

In the manuscript, the authors evaluated the effect of PTSO and PTS on many tumor cell lines, but the number and the type of cell lines differed in each experiment. The antiproliferative effect (IC50) was evaluated on 9 tumor cell lines and on PBMCs as model of normal cells; the DNA damage assay was performed on MCF-7, the oxidative stress assay on MCF-7 and T-84 (in this case ROS production was confirmed only on MCF-7 using DCFH-DA), the analysis of apoptosis  on MCF-7 and T-84, the anti-inflammatory properties on HT-29 and T84. Since the toxicity of other OSCs is independent by ROS production in some type of tumor cells, to elucidate the cytotoxicity mechanism  of PTSO and PTS, the number of cell lines used in the experiments should be increased. Furthermore, if the authors will choose to use different cell lines in different experiment, they should be explain their choice.

Minor points

Considering the text content, references 3 and 4 should be replaced with more pertinent ones.

Line 232- the p value for the statistical significance is incorrect

Line 237- “Both compounds inhibited cellular proliferation …with different intensity against….the word “intensity “ is not correct, should be changed.

In the results of analysis of apoptosis the authors distinguished between early and late apoptosis, but it is not reported at which time these analysis were performed

Fig.8, Fig.9 and Fig.10. “ Different letters between columns indicate significant differences at p <0.05” This way of showing the results of statistical analysis is not clear . In the legend should be specify each the correspondence of each letter or the graphical representation changed

Author Response

Reviewer#2. Comments and Suggestions for Authors

-This manuscript is focused on anti-proliferative, pro-apoptotic and anti-inflammatory activities of two OSCs obtained by Allium spp, PTSO and PTS,  in several tumor cell cultures.  The in vitro experiments revealed that these bioactive compounds can reduce cell proliferation and induce apoptosis and indicate ROS production as one of potential mechanisms underlying their cytotoxicity effect. Furthermore, the anti-inflammatory effect was investigated by assayed the pro-inflammatory cytokine production.  The topics are well-presented, the language is clear and the text is well structured. The novelty of the work is interesting and the rational is clear, but there are several limitations in the results, then the manuscript is still a long way from publication.

COMMENTS FOR THE AUTHOR:

Major points:

-In the manuscript, the authors evaluated the effect of PTSO and PTS on many tumor cell lines, but the number and the type of cell lines differed in each experiment. The antiproliferative effect (IC50) was evaluated on 9 tumor cell lines and on PBMCs as model of normal cells; the DNA damage assay was performed on MCF-7, the oxidative stress assay on MCF-7 and T-84 (in this case ROS production was confirmed only on MCF-7 using DCFH-DA), the analysis of apoptosis  on MCF-7 and T-84, the anti-inflammatory properties on HT-29 and T84. Since the toxicity of other OSCs is independent of ROS production in some types of tumor cells, to elucidate the cytotoxicity mechanism  of PTSO and PTS, the number of cell lines used in the experiments should be increased. Furthermore, if the authors choose to use different cell lines in different experiments, they should explain their choice.

 Response: The work aimed to evaluate the antitumoral effect of two compounds. In the first approximation, we evaluated the antiproliferative effect in different cell lines (MCF-7; T-84; A-549; Ht-29; Panc-1; Jurkat; SW-837; PC-3; T1-73; PBMCs). Later, specific experiments were performed in particular cell lines that haven been previously used for those determinations. 

Minor points

-Considering the text content, references 3 and 4 should be replaced with more pertinent ones. 

Response: We thank the reviewer's comment, the references have been replaced. 

-Line 232- the p value for the statistical significance is incorrect.

Response: We apologize for the mistake, the manuscript has been widely revised. 

-Line 237- “Both compounds inhibited cellular proliferation …with different intensity against….the word “intensity “ is not correct, should be changed.

Response: We appreciate the suggestion, and consequently, the sentence has been changed. 

-In the results of analysis of apoptosis the authors distinguished between early and late apoptosis, but it is not reported at which time these analysis were performed.

Response: An analysis of the apoptotic stages and necrotic cells has been included. 

-Fig.8, Fig.9 and Fig.10. “ Different letters between columns indicate significant differences at p <0.05” This way of showing the results of statistical analysis is not clear. In the legend should be specify each the correspondence of each letter or the graphical representation changed

Response: We appreciate the reviewer comment. However, we usually use different letters to show statistically significant differences between groups when many different groups are evaluated and when we want to compare all groups with each other. Thus, in columns/groups with the same letter, the difference between the means is not statistically significant (p<0.05). However, if two columns/groups have different letters, they are significantly different (p<0.05).

Round 2

Reviewer 1 Report

This study is deserved to be published in Nutrient.

Author Response

We highly appreciate the reviewer comments

Reviewer 2 Report

The author’s responses are answers are quite satisfactory, but there is another point :

 In the DNA damage assay microscopic images of 8-OHdG production do not show a significant difference between cells treated with PTS and PTSO and controls with values close to 0. The authors did not comment this result and only show the fluorescence of DAPI. In general, DAPI is used as fluorescent dye for focusing and quality evaluation. Observing the pictures, it is easy to notice that the difference in intensity of fluorescence is due to the different number of cells that, on the contrary, should be very similar to allow this evaluation. Although this compounds produce cell damage and apoptosis, this assay was performed after 1 h of treatment with PTS and PTSO (as reported in the materials and methods). The authors reported that this treatment is 2h in the discussion.

However, the values fluorescence of 8-OHdG-FITC antibody are too low to allow a correct comparisons of PTS and PTSO treated cells to the control and it is possible to see only a fluorescent cell in  PTSO-treated sample

In conclusion, the authors can not use these images to demonstrate that PTS and PTSO produced DNA damage. If it is impossible find new images , they can choose to eliminate this paragraph and change the discussion.

Author Response

We thank the careful and insightful review of this manuscript. Thus, following the reviewer suggestion, DNA damage determination has been removed from the revised manuscript.